# Impact of Bed Form Celerity on Oxygen Dynamics in the Hyporheic Zone

**Philipp Wolke** [1,2], **Yoni Teitelbaum** [3], **Chao Deng** [3], **Jörg Lewandowski** [1,4] **and Shai Arnon** [3,*]

1   Department Ecohydrology, Leibniz-Institute of Freshwater Ecology and Inland Fisheries, Müggelseedamm 310, 12587 Berlin, Germany; philipp.wolke@web.de (P.W.); lewe@igb-berlin.de (J.L.)
2   Institute of Geological Sciences, Department of Earth Sciences, Freie Universität Berlin, Malteserstr. 74-100, 12249 Berlin, Germany
3   Zuckerberg Institute for Water Research, The Jacob Blaustein Institutes for Desert Research, Ben-Gurion University of the Negev, 84990 Midreshet Ben-Gurion, Israel; ytbaum@gmail.com (Y.T.); marschao@163.com (C.D.)
4   Geography Department, Humboldt University Berlin, Rudower Chaussee 16, 12489 Berlin, Germany
*   Correspondence: sarnon@bgu.ac.il; Tel.: +972-8-6563-510

**Abstract:** Oxygen distribution and uptake in the hyporheic zone regulate various redox-sensitive reactions and influence habitat conditions. Despite the fact that fine-grain sediments in streams and rivers are commonly in motion, most studies on biogeochemistry have focused on stagnant sediments. In order to evaluate the effect of bed form celerity on oxygen dynamics and uptake in sandy beds, we conducted experiments in a recirculating indoor flume. Oxygen distribution in the bed was measured under various celerities using 2D planar optodes. Bed morphodynamics were measured by a surface elevation sensor and time-lapse photography. Oxygenated zones in stationary beds had a conchoidal shape due to influx through the stoss side of the bed form, and upwelling anoxic water at the lee side. Increasing bed celerity resulted in the gradual disappearance of the upwelling anoxic zone and flattening of the interface between the oxic (moving fraction of the bed) and the anoxic zone (stationary fraction of the bed), as well as in a reduction of the volumetric oxygen uptake rates due shortened residence times in the hyporheic zone. These results suggest that including processes related to bed form migration are important for understanding the biogeochemistry of hyporheic zones.

**Keywords:** hyporheic exchange; bed form migration; moving streambed; ripples; planar optodes

## 1. Introduction

The hyporheic zone (HZ) plays a major role in various physical, chemical, and biological processes in streams and rivers [1–3]. The HZ is often characterized as the zone in which flow paths of stream water enter the sediment, and re-emerge back to the stream after spending some time in the subsurface [4]. Water flux through the HZ is termed hyporheic exchange flux (HEF). The mixing of stream water and groundwater occurs in the HZ, and this may change the physical properties (e.g., temperature) and chemical composition of the water. Such mixing leads to steep chemical gradients, and therefore, the HZ is a highly-reactive environment with intense chemical turnover rates, which are catalyzed by microorganisms [2,5–7]. This includes nutrient cycling, metal transformation, and contaminant degradation [8–10]. Understanding the processes in the HZ, therefore, has implications for water resources management and stream restoration [11,12].

Dissolved oxygen is the key solute driving the metabolism of organisms living in rivers and within the HZ. The consumption rate of oxygen by microorganisms is an indicator of biochemical turnover rates, and dictates the redox zonation in the streambed.

The flux of oxygen across the sediment–water interface and the size of the oxygenated zone is strongly regulated by bed topography, stream water velocity, and oxygen consumption rate [13,14]. Therefore, variations of the flow regime, river morphology, and the regional groundwater flow pattern alter the exchange across the sediment–water interface, and thus, also the extent of the HZ [15,16]. Because HEF in sandy rivers is primarily induced by bed forms, their ubiquitous occurrence makes them the most important contributor to HEF and associated biogeochemical processes [17]. The driving forces of flow through the HZ are the hydraulic head variations at the immediate sediment–water interface, which are developed by water flowing over bed forms. Solute transport by advective flow due to the head difference along the bed forms is often called "advective pumping" [18], and is often used to model transport processes in sandy sediments.

Sandy and silty streambeds are highly sensitive to shear forces and are mobile under flow conditions that are commonly found in streams and rivers. The motion of the streambed can be characterized by the speed of its movement (i.e., celerity). During the movement of the bed, sediment particles roll from the stoss side and avalanche down the lee side of the bed form. During such sand movement, the trapping and release of solute occur as bed forms propagate, and contribute to the exchange by advection. This physical exchange mechanism is termed "turnover" [18]. During bed form motion, advective pumping and turnover can occur simultaneously, and their relative contributions result in altered flow paths and residence time distribution of pore water as compared to pure advection under stationary bed conditions. With increasing bed form celerity, turnover becomes more dominant as compared to advective pumping. Ultimately, when the bed form celerity is greater than the pore water velocity, the penetration of solutes is mostly restricted to the extent of the moving bed forms, and the exchange between the sediment layer affected by bed form celerity and the immobile sediment layer underneath is primarily limited to dispersion and diffusion. Thus, bed form celerity can have a major influence on the distribution and residence time of water entering the sediment, as well on oxygen availability for microorganism and redox processes [19–21]. However, despite the potential impact of moving streambeds on ecological and biogeochemical processes, there is a scarcity of such studies, because studying moving streambeds is much more challenging than studying stagnant streambeds.

Oxygen dynamics under moving bed form conditions have so far only been considered in a few modelling studies [19,21,22], and in two experimental studies in marine systems [23,24]. The only experimental study on a freshwater system focused on mobile dunes, several meters in size [25]. To the best of our knowledge, controlled experiments have never been conducted in a freshwater system. The main objective of the present study is to quantify the effect of bed form celerity on oxygen dynamics and oxygen consumption in moving bed forms.

## 2. Materials and Methods

### 2.1. Experimental Setup, Conditions, and Approach

The experiments were conducted in a recirculating indoor flume with a working section dimension of 260 cm in length and 29 cm in width, as described in detail by De Falco et al. [26]. A variable-speed pump controlled the water flow in the flume channel, while the discharge was measured by a magnetic flow meter (SITRANS F M, MAG 5100 W Siemens, Nordborg, Denmark) integrated in the return pipe between pump and flume inlet. Water in the flume was kept between 24 and 25 °C by a chiller (TR/TC 10, TECO Refrigeration Technologies, Ravenna, Italy). Natural sandy sediment from the Yarqon River in central Israel was used in the experiments [27]. The sediment was excavated from the upper 10 cm of the streambed, and was taken directly to the laboratory. The sediment was kept moist in ambient air for three days, during which it was mixed several times. Afterwards, the sediment was wet-sieved to remove all particles larger than 2 mm and was added to the flume channel, which was filled with

deionized water. Flow was initiated soon after the sediment was added to the channel. The recirculating system carried all the sediment washed from the outlet of the channel, and reintroduced it through the inlet so that the net amount of sediment in the flume would remain unchanged.

A total of eleven experiments were performed with different flow conditions (Table 1). Each flow condition was duplicated, except for the streamflow velocity of 0.32 m·s$^{-1}$, which was repeated three times. The duplicates were labeled Set 1 and Set 2, respectively, to elucidate possible temporal changes, and because true replication is not possible when using such large experimental systems. Each experiment was centered around the measurement of oxygen distribution in the sediment, which was conducted by planar optodes. Each batch of oxygen measurements took a few hours, with images captured every 5–15 min. Bed form morphodynamics were quantified using time-lapse digital photographs to quantify the bed form length and celerity, and depth tracking with an acoustic profiler to quantify the bed height. Temperature, pH, oxygen saturation, and electrical conductivity (EC) were measured in the stream water with a handheld meter (WTW, Multi 3320, Weilheim, Germany) before and after oxygen distribution measurements. A waiting time of approximately 24 h was implemented before the next run started, letting the system adapt to the new flow conditions, which was defined as a stable distribution of oxygen within the sediments. HEF was measured under all stream water velocities after finishing all oxygen measurements. Afterwards, sediment samples were collected from the mobile and immobile section of the bed. Porosity, organic matter content, and grain size distribution were determined based on these sediment samples. Hydraulic conductivity was calculated from the particle size distribution using HydrogeoSieveXL and the Hazen equation [28].

### 2.2. Bed Form Morphodynamics and Sediment Characterization

Bed form height was evaluated with a Doppler velocimeter (Nortek Vectrino II Profiler, Rud, Norway), which was fixed in a single known position, and by using the depth tracking mode to measure the distance to the bed as the bed moved. The bed topography dynamics were logged for up to six days under the slowest celerity, and two days under all other celerities, to ensure that the number of moving bed forms was sufficient for statistical analysis (Table 1). The time series were evaluated with the "find peaks" method, in which bed form troughs are represented by local minima and crests by local maxima. A minimum required prominence level of 5 mm was set in order to exclude small height variations on larger bed forms, which would otherwise be identified as individual bed forms themselves. The mean bed height was calculated using the arithmetic mean of the recorded bed form troughs and crests. The final bed form height of each condition was determined as the average of the data of the two or three experimental runs, respectively.

The 1-dimensional bottom tracking measures could not be used to calculate the bed celerity. Therefore, bed form celerity was measured with a digital camera (Nikon D5300, with AF-P DX NIKKOR 18–55 mm VR Lens, Ayuthaya, Thailand) by taking images at the same place as the optode, and simultaneously with oxygen imaging in order to ensure correct coupling between the oxygen distribution and the bed form shape (Appendix A, Figure A1). The bed form celerity during the experiments was determined by using an ad-hoc Python code. This software made it possible to track the pixels of bed form crests and troughs on the recorded images over several images. The average position change over time of these pixels was translated into the celerity.

Porosity was measured by filling a container with a known volume with saturated sediment (100 mL) and measuring the loss of weight after drying at 105 °C for two days. Organic matter content was measured by loss through ignition after burning a 5 g sediment sample at 450 °C for 4 h. The methods followed the protocols shown in Klute et al. [29].

### 2.3. Hyporheic Exchange Flux

A total of 19 tracer experiments were carried out to determine HEF under different flow conditions. In these experiment, 100 g NaCl was added to the surface water, which increased the EC of the overlying water from ~400 to ~900 μS·cm$^{-1}$. The EC in the overlying water was recorded every 5 s,

and HEF was calculated from the initial decline of the EC due to hyporheic exchange, as described by Packman et al. [30] and Fox et al. [31]. Briefly, the interactions between the streamflow and the sediment produced pressure head variations over bed forms, with high pressure zones on the stoss side and low pressure zones on the lee side of the bed forms.

The outcome of this is a two-dimensional advective flow field within the sediments, with water flowing into (on the stoss side), through, and then out of the bed (on the lee side). Shortly after the addition of the tracer NaCl to the stream water, the EC of the stream water was higher than that of the pore water. Thus, HEF delivered high-EC water into the bed while porewater with lower EC returned to the stream and diluteed the stream water, i.e. reduceed the EC. Solving mass balance equations allowed us to extract the exchange flux from the initial slope of the EC decline, as explained in detail by Packman et al. [30] and Fox et al. [31].

*2.4. Oxygen Imaging with Planar Optodes*

The distribution of oxygen in the sediment was measured with a planar optode system (VisiSensTD, Presens GmbH, Regensburg, Germany). An optode with a size of $10 \times 15$ cm was attached to the inner glass side wall of the flume such that it captured both the streambed and its interface with the stream water (Appendix A, Figure A1). The basic physical principle of oxygen sensing with planar optodes is the dynamic quenching of a luminescence indicator substance in the presence of oxygen. The method has great advantages over conventional oxygen measurements, since it does not consume oxygen, and allows non-invasive 2D imaging of oxygen in the bed to be undertaken with high precision [32]. The flume was covered with black cloth during the entire duration of the experiment, since the fluorescent substances in the optode are photosensitive and degrade with increased exposure to light. Light was used only for short periods when photographs were taken.

*2.5. Data Analysis*

Image processing techniques were used to combine the topographic information from the digital image of the sediment–water boundary with the spatial oxygen saturation information of the calibrated optode images. This allowed us to exclude areas of oxygenated surface water from the optode images. In order to use the optode images, it was assumed that they were representative of the porous medium, i.e., that no significant wall effects occurred. The images were batch processed using an ad-hoc Python code with the open source computer vision library OpenCV [33]. A flow diagram of the processing steps is given in Appendix A (Figure A2).

The preprocessed images allowed us to determine the size and mean oxygen saturation of the oxygenated area at high resolution (pixel size was 15.625 $\mu m^2$). The oxygenated zone was defined here using a threshold value of 15% oxygen saturation. The size of oxic zone $A_{ox}$ was calculated by Equation (1):

$$A_{ox} = A_{pix} \sum_{i=1}^{n} P_i \tag{1}$$

where pixel $P_i$ is 1 if it is located in the streambed and the oxygen saturation is above the threshold (otherwise, $P_i$ is 0), $A_{pix}$ is the area of one pixel (15.625 $\mu m^2$), and n is the total number of pixels of the planar optode. The mean oxygen concentration of the oxic zone $C_{oxz}$ is calculated by Equation (2):

$$C_{oxz} = \frac{1}{n} \sum_{i=1}^{n} P_i \cdot C_i \tag{2}$$

where $C_i$ is the oxygen concentration of the individual pixel.

The volumetric oxygen consumption rates $R$ (mg $l^{-1} \cdot h^{-1}$) of the pore water were calculated with two different methods: the "maximum uptake method", and the "delta method". Both methods are based on averaging the flux and oxygen concentrations within the two-dimensional porous domain,

and thus, represent more reliable results than previously used 1-dimensional approaches [34,35]. We refer to this process in the text as oxygen uptake. The first method, suggested by Ahmerkamp et al. [21], is referred to as the maximum uptake method, $R_{max}$, represented with Equation (3):

$$R_{max} = \frac{F_{\downarrow O_2}}{\delta \times \theta \times 24\,h} \tag{3}$$

where $\theta$ is the porosity, $F_{\downarrow O_2}$ (Equation (4)) represents the flux of oxygen into the bed:

$$F_{\downarrow O_2} = C_{sw} \times q_h \tag{4}$$

and the mean oxygen penetration depth $\delta$ is calculated with Equation (5):

$$\delta = \frac{A_{ox}}{W_{opt}} \tag{5}$$

where $C_{sw}$ is the oxygen concentrations in the stream water, $q_h$ is the HEF, and $W_{opt}$ is the optode width. In this case, it was assumed that all oxygen transported from the stream water into the sediment by HEF was consumed within the sediment, and that no oxygen had returned to the stream water by upwelling flow paths.

The second method is referred to as the delta method (Equation (6)).

$$R_{delta} = \frac{F_{\downarrow O_2} - F_{\uparrow O_2}}{\delta \times \theta \times 24\,h} \tag{6}$$

where $F_{\uparrow O_2}$ ( Equation (7)) represents the flux of oxygen out of the bed:

$$F_{\uparrow O_2} = C_{oxz} \times q_h \tag{7}$$

The delta method is based on the assumption that the HEF that enters the subsurface with flux $q_h$ and oxygen concentration $C_{sw}$ reemerges to the surface water with the same $q_h$ but with an altered oxygen concentration due to the microbial consumption. For brevity, the mean oxygen concentration in the oxic zone ($C_{oxz}$) is used.

In addition to the experimentally-determined HEF, the HEF due to pumping was calculated following Elliott and Brooks [18] theory by using Equation (8).

$$\bar{q} = \frac{K \times k \times h_m}{\pi} \tag{8}$$

where $\bar{q}$ refers to spatially averaged flux into the bed, $K$ is hydraulic conductivity, $k$ is the wavenumber (given by the equation $k = 2\pi/\lambda$, where $\lambda$ is the bed form wavelength), and $h_m$ is the head variation over the bed form (see more details in Elliott and Brooks [18]).

Finally, the flushing time of the bed was calculated following the definition of Monsen et al. [36], which is an integrative approach consisting of taking the oxygenated pore water volume of the hyporheic zone ($A_{ox}$ multiplied by the flume width and $\theta$) and dividing by the volumetric flux through the oxygenated volume (i.e., HEF multiplied by the flume width and optode width, $W_{opt}$). This only serves as an approximation of time it takes to flush the oxygenated volume despite the fact that flow paths in the streambed has variable lengths and that water velocities are lognormally distributed [14]. The concept of flushing time is widely used in much of the hyporheic zone literature, although in many cases it is referred to as residence time [37,38].

## 3. Results

### 3.1. Water and Sediment Characteristics

The turbidity of the surface water increased with stream water velocity, from 62 NTU under stationary conditions up to 478 NTU under the maximum stream velocity of 0.37 m·s$^{-1}$ (Table 1). EC was also not constant, and increased gradually from 291 μS·cm$^{-1}$ to 362 μS·cm$^{-1}$ during the two weeks of the experiments. pH and oxygen were relatively stable, ranging between 8.1–8.2 and 92.8% to 98.7% for pH and oxygen, respectively.

Comparisons between the sediment in the moving fraction (upper few cm) and the immobile fraction below revealed that the mobile fraction had a slightly higher porosity (36% vs. 33.6%), but that these differences were not significant (t-test, $p > 0.05$). This is also true for the median grain diameters, which were 0.264 mm and 0.298 in the moving fraction and immobile fraction, respectively. The hydraulic conductivity was calculated based on the grain size distribution, and was found to be almost the same for the moving and immobile fractions ($4.62 \times 10^{-4}$ m·s$^{-1}$). Significantly higher organic matter content was observed in the immobile fraction as compared to the mobile sediment layer (1.38% vs. 1.05%, t-test $p < 0.05$).

The calculated bed form celerities and mean heights of bed forms under each experimental run are given in the Table 1. It was observed that the bed form celerity under the same flow conditions could be temporally variable, while some bed forms tended to accelerate or decelerate at certain time points without clear explanation. This is expressed in the increased standard deviation under higher celerities (e.g., Table 1 and Supplementary Material, Figures S1–S9 and Movies S1–S9).

It was also observed that suspended fine material deposits resulted in a very thin layer at the interface between the bed and the stream water when the bed forms were stationary (slowest flow conditions). The mean bed form length was 13.3 (± 3.8) cm, but there was no clear trend with celerity. Finally, the mean height of the bed forms increased from 1.13 cm to 1.37 cm between the slowest and intermediate celerity (0.04 m·h$^{-1}$ and 0.14 m·h$^{-1}$). However, this is increase in mean height was somewhat smaller than the variation in height within each flow condition, which can be exemplified with the observed standard deviations that ranged between 0.42 and 0.60 cm.

**Table 1.** Physical and chemical properties of the water and sediment during the experiments.

| Run No./Set No. | Stream Water Velocity (m·s$^{-1}$) | Bed Form Height (cm)[1] | Bed Form Celerity (m·h$^{-1}$)[1] | Water Depth (cm) | Temp. (°C) | EC (μS·cm$^{-1}$) | Turbidity (NTU) |
|---|---|---|---|---|---|---|---|
| 1/1 | 0.16 | 1.50 (N/A) | 0.000 (N/A) | 14.2 | 24.9 | 291 | 62 |
| 7/2 | 0.16 | 1.94 (N/A) | 0.000 (N/A) | 14.2 | 24.3 | 339 | 189 |
| 3/1 | 0.25 | 1.13 (0.42) | 0.035 (0.000) | 13.8 | 24.9 | 322 | 126 |
| 8/2 | 0.25 | 1.13 (0.42) | 0.049 (0.000) | 13.7 | 24.1 | 345 | 137 |
| 4/1 | 0.28 | 1.37 (0.51) | 0.140 (0.004) | 14.2 | 24.4 | 326 | 141 |
| 9/2 | 0.29 | 1.37 (0.51) | 0.135 (0.006) | 13.9 | 24.8 | 355 | 308 |
| 2/1 | 0.32 | 1.38 (0.46) | 0.394 (0.110) | 14.2 | 24.5 | 299 | 267 |
| 5/1 | 0.32 | 1.38 (0.46) | 0.275 (0.045) | 14.2 | 24.4 | 331 | 230 |
| 10/2 | 0.33 | 1.38 (0.46) | 0.375 (0.105) | 13.9 | 24.2 | 359 | 375 |
| 6/1 | 0.36 | 1.41 (0.60) | 0.699 (0.287) | 14.3 | 24.4 | 335 | 311 |
| 11/2 | 0.37 | 1.41 (0.60) | 0.644 (0.080) | 13.8 | 24.6 | 362 | 478 |

[1] standard deviations shown in parenthesis.

### 3.2. Dynamics of HEF and Oxygen Distribution

HEF increased monotonically from the slowest stream water velocity (stationary streambed) until it reached a maximum at stream water velocity of about 0.35 m·s$^{-1}$ (celerity of 0.35 m·h$^{-1}$). Further increase in stream water velocity resulted in a decrease of HEF (Figure 1). The variability among the different experiments with the same flow conditions was attributed mainly to experimental

sensitivity, which was larger at faster stream water velocities. Exchange fluxes were modeled to calculate predictions based on advective pumping [18]. Modelling was conducted using a hydraulic conductivity of $4.62 \times 10^{-4}$ m·s$^{-1}$. For stationary beds, the modelled results were similar to the measurements, but deviations increased as celerity increased (Figure 1).

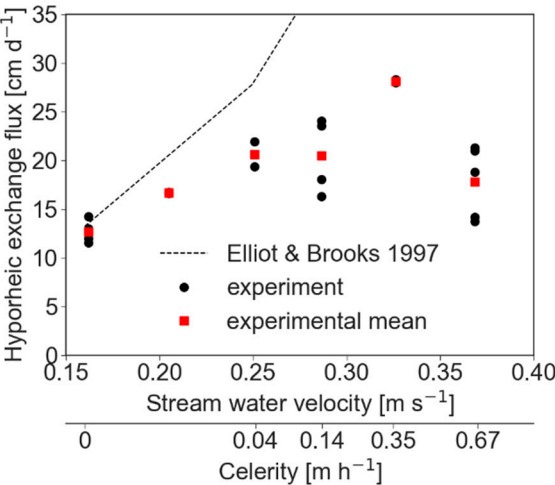

**Figure 1.** Relationship between HEF and stream water velocity and bed form celerity (bed form celerity axis is not scaled). The dashed line shows modelled results for stationary bed forms by including only the effect of pumping. The predicted exchange continuously increases in a linear manner until it reaches 74 cm·s$^{-1}$ under the stream water velocity of 0.37 m·s$^{-1}$ (data not shown).

The oxygenated zone under stationary bed conditions had a typical conchoidal shape (Figure 2A). A distinctly developed upwelling zone of anoxic water at the bed form trough was observed under stationary conditions, but gradually diminished until it disappeared under faster velocities (Figure 2). These upwelling zones of deeper pore water with low oxygen concentrations are often termed "chimneys" [21]. When these chimneys disappeared at higher celerities, the moving fraction of the bed forms were usually well oxygenated, and a thin transition zone with a steep oxygen gradient towards the deeper parts of the bed were very typical (Figure 2 and Supplementary Material, Figures and Movies S5–S9).

It was also observed that as the bed forms moved, the chimneys disappeared and reappeared when the next bed form passed through the optode (Supplementary Material, Figures S2: image 3–8 and S4: image 1–8, and Movies S2, S4). However, above a celerity of 0.35 m·h$^{-1}$, the chimneys were no longer observed, and the interface between the oxic and anaerobic zone became flatter, as compared to slower celerities. It is important to mention that the oxygen scale in Figure 2 slightly exceeds 100% oxygen saturation (bright color). This stems from the technical inability to provide the same lighting to the whole optode. Because light is focused more in the center, deviations are usually observed in the edges of the planar optodes. Additionally, planar optodes are less precise at high oxygen saturation levels compared to lower ones. Corrections of the surface water oxygen saturation could be done easily, since we measured oxygen saturation in the stream water using an electrode. It was found that the mean oxygen saturation of the pixels above 100% of all experimental runs was 107.3% ± 1.1%, and the area they occupied was 8.9% ± 7.4%. The high standard deviation was caused by larger proportions of these pixels at celerities above 0.35 m·h$^{-1}$.

The size of the oxygenated zone increased dramatically right after the bed forms started to move, but a more modest increase in its size was observed as velocity increased further (Figure 3). For example, during stationary conditions, the oxygenated areas in the two runs were 23.39 and 31.23 cm$^2$, as compared to 53.73 and 41.79 cm$^2$ under slow celerity of 0.04 m·h$^{-1}$. Interestingly, the first runs of all experimental conditions show less oxygenated sediment in the subsurface than second runs,

with the slow celerity experiment being an exception (Figure 3B). Moreover, the size of the oxygenated area was positively linked to turbidity (Figure 4).

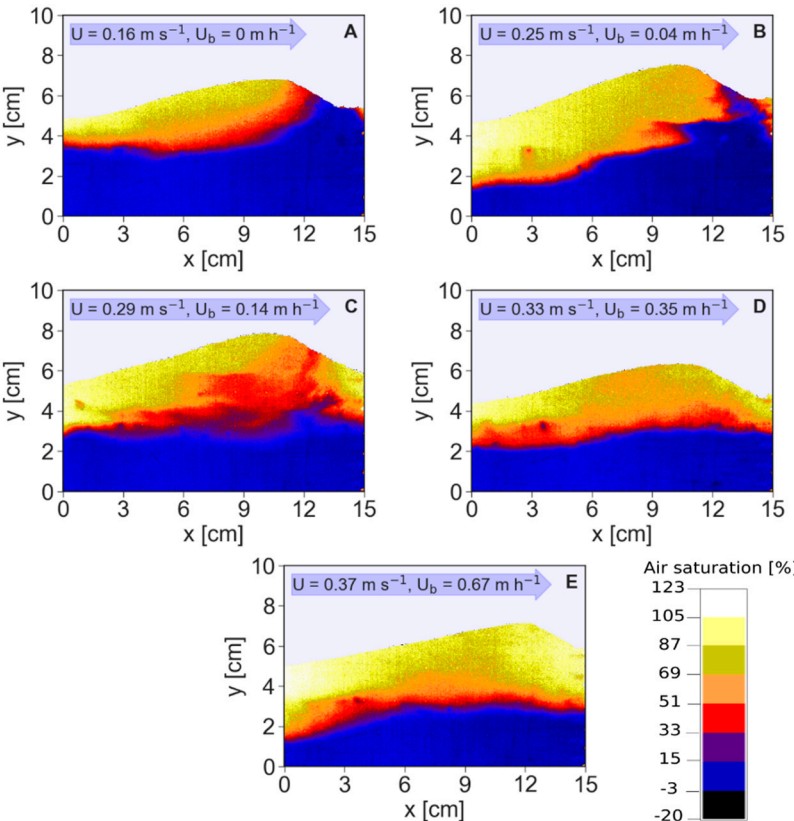

**Figure 2.** Optode images showing spatial oxygen distributions in the streambed under increasing streamflow velocities (U) and bed form celerities ($U_B$), from stationary conditions to the fastest bed form celerity. Streamflow velocities (U) and bed form celerities ($U_B$) are shown on Panels (**A**–**E**). Images and videos of the other flow conditions are given in the Supplementary Material.

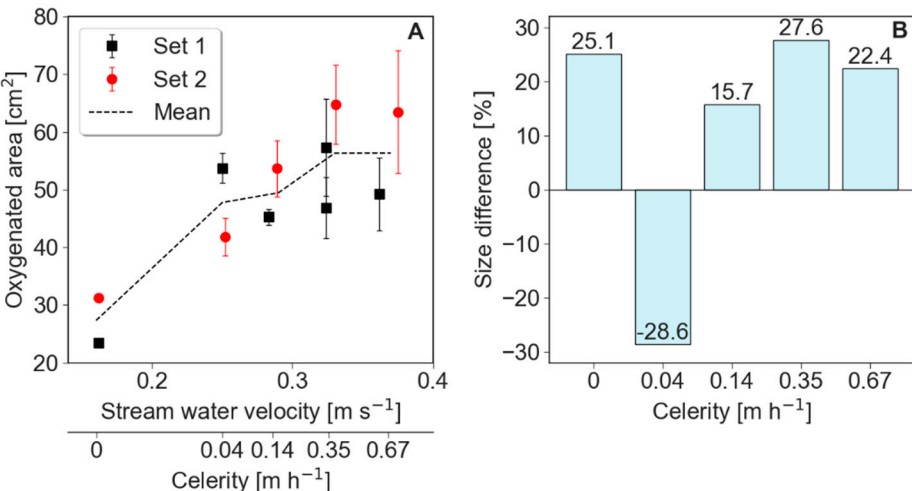

**Figure 3.** The spatial and temporal mean of the total oxygenated area under different stream water velocities and celerities (**A**). Each condition is displayed chronologically as Set 1 and 2, and the dotted line is the arithmetic mean. Error bars denote standard deviations of each measurement. In general, the areas in Set 2 were greater than in Set 1, except for bed form celerity of 0.14 m·h$^{-1}$, as shown by the percentage difference in area size between Set 1 and 2 (**B**).

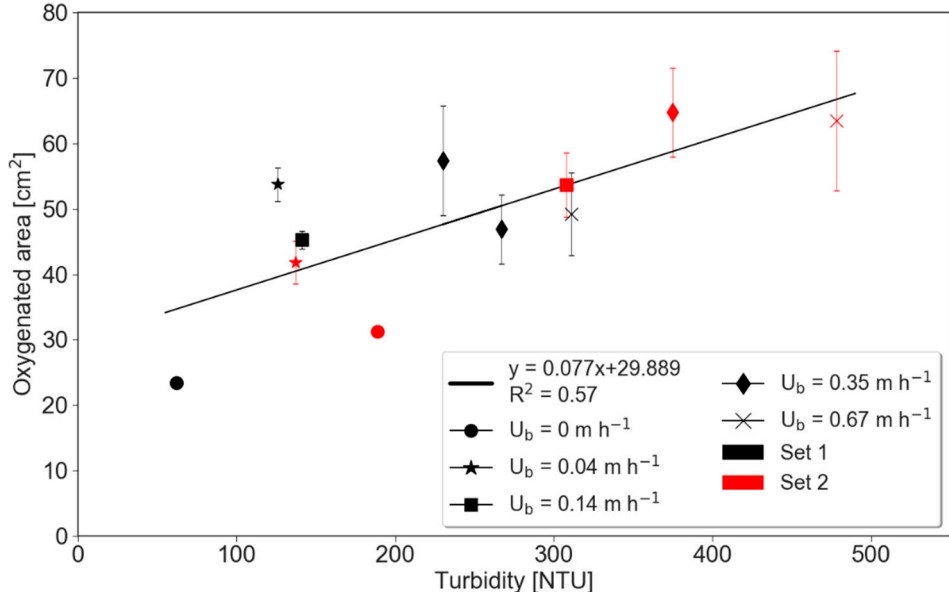

**Figure 4.** Relationship between oxygenated area and surface water turbidity. The experimental conditions, labelled by their celerities, are additionally separated into Set 1 and 2 (first and second run of the same celerity). Whiskers denote standard deviations.

*3.3. Oxygen Fluxes and Uptake Rates*

The oxygen flux into the sediment was derived with Equation (4) using the mean of the HEF, as shown in Figure 1, and the oxygen concentrations in the water (Table 1). The calculated oxygen fluxes following Equation (4) trailed the pattern of the HEF (Table 2). Table 2 also displays modeled fluxes following the model presented by Ahmerkamp et al. [20]. Nevertheless, although the prediction following Ahmerkamp et al. were of the same order of magnitude (and only slightly lower for stationary bed forms), increasing deviations were observed as celerity increased. Oxygen uptake rates that were calculated with the delta method covered a range from 22 to 75.1 $\mu$mol$\cdot$L$^{-1}\cdot$h$^{-1}$, and were considerably lower than rates of the maximum uptake method with 125.1 to 252.4 $\mu$mol$\cdot$L$^{-1}\cdot$h$^{-1}$ (Figure 5). The oxygen uptake rates showed a decreasing trend with increasing stream water velocity and associated bed form migration celerity in both model approaches. The aforementioned shift of the size of the total oxygenated area between experimental runs (Figure 3) translates into higher oxygen uptake rates of the experimental results from Set 1. An exception is again the first run under the slow celerity (0.04 m$\cdot$h$^{-1}$), Run 2, which was the first measurement with a moving bed. It exhibits an unusually high value, but is not included in calculating the trends in Figure 5A,B (marked as an outlier in the legend). Because HEF, oxygen fluxes, and the oxygenated zone were all influenced by bed form celerity in a complex manner, the resulting oxygen consumption rates varied as well, and were negatively correlated with increasing flushing times (Figure 5C,D).

**Table 2.** Calculated oxygen flux into the sediment (Equation (4)), modelled oxygen influx based on the equations provided by Ahmerkamp et al. [20], and calculated oxygen flux out of the sediment based on the assumption underlying the delta method (Equation (7)).

| Run No./Set No. | Oxygen Influx (mmol$\cdot$m$^2\cdot$d$^{-1}$) [1] | Oxygen Outflux (mmol$\cdot$m$^2\cdot$d$^{-1}$) [1] | Modelled Oxygen Influx (mmol$\cdot$m$^2\cdot$d$^{-1}$) |
|---|---|---|---|
| 1/1 | 28.48 (2.32) | 20.07 (1.64) | 14.67 |
| 7/2 | 29.59 (2.41) | 18.98 (1.55) | 10.55 |
| 3/1 | 47.48 (2.96) | 34.69 (2.16) | 11.09 |
| 8/2 | 48.11 (2.99) | 34.92 (2.18) | 15.66 |
| 4/1 | 47.91 (7.92) | 34.74 (5.75) | 14.43 |

**Table 2.** *Cont*.

| Run No./Set No. | Oxygen Influx (mmol·m²·d⁻¹) [1] | Oxygen Outflux (mmol·m²·d⁻¹) [1] | Modelled Oxygen Influx (mmol·m²·d⁻¹) |
|---|---|---|---|
| 9/2 | 47.50 (7.85) | 33.02 (5.46) | 11.69 |
| 2/1 | 65.01 (0.38) | 45.84 (0.27) | 21.32 |
| 5/1 | 65.80 (0.39) | 49.13 (0.29) | 17.00 |
| 10/2 | 65.73 (0.39) | 50.69 (0.30) | 14.48 |
| 6/1 | 42.48 (7.78) | 31.43 (5.75) | 12.16 |
| 11/2 | 42.91 (7.86) | 35.52 (6.50) | 9.11 |

[1] standard deviations shown in parenthesis.

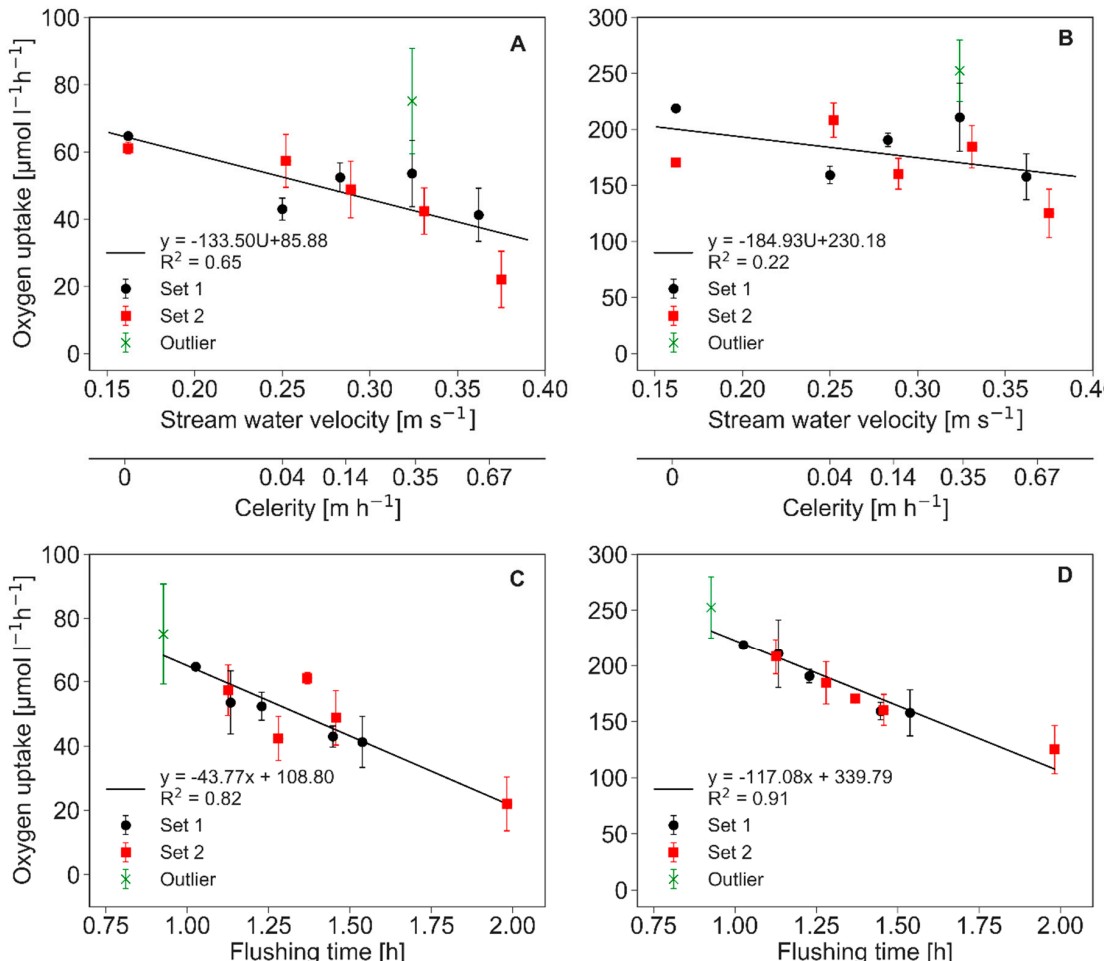

**Figure 5.** Oxygen uptake rates under different stream water velocities, bed form celerities, and flushing times of the oxygenated zone. The delta method (**A**) and (**C**) is based on the assumption that a proportion of oxygen in the hyporheic zone is transported back to the surface water. The maximum uptake method (**B**) and (**D**) is based on the assumption that all oxygen transported into the subsurface is consumed. The standard deviations are shown as error bars.

## 4. Discussion

### 4.1. Bed Form Morphodynamics and Flow

The sizes and shapes of the bed forms that were captured during the oxygen measurements were relatively similar under the different flow conditions (Table 1 and Supplementary Material, Figures and Movies S1–S9), i.e., similar to bed forms reported in other studies [39,40], and similar to bed forms

predicted by models [22]. However, the amount of captured bed forms could not cover the whole range of variability that was expected, since previous studies have suggested that tens of bed forms are required to statistically characterize their variability. Although bed form morphodynamics were extensively studied in the past, efforts were made to collapse their statistical behavior into simple scaling laws, rather than to discuss their variability [41]. Consequently, it has been assumed in many biogeochemical studies that bed form shapes are unchanging, even though they are constantly in motion. Thus, they were modelled using a frame that moves at the bed form celerity [21,22]. Using this approach did not include temporal dynamics such as those shown here.

In addition, some bed parameters changed due to grain sorting processes, which slightly changed the grain size and porosity, and significantly reduced the organic matter content in the mobile fraction, as compared to the non-mobile fraction. It was also observed that the mean height of the bed forms slightly increased with increasing stream water velocity, which is common behavior in sandy streambeds [42,43]. The most prominent difference that was observed under different velocities was the change of the bed form celerity, which occurred due to the change in the shear forces over the bed. Despite the fact that bed form celerity was not calculated over prolonged periods that are sufficient to establish statistical distributions of the temporal movement dynamics, the observed celerities were paired to the measurements of oxygen uptake (Figures 3 and 5), thus making it possible to link celerity to oxygen uptake. In addition, the trend of celerity increase with increasing velocities was similar to the trends observed in previous experimental studies with sand [20,39].

HEF increased nearly linearly with velocity until reaching a maximum value at a celerity of $0.35 \, \text{m·h}^{-1}$. This trend is different from what was seen under slower streamflow velocities with stationary bed forms, where HEF increases proportionally to the square of the stream water velocity [44,45]. This change in the trend towards a linear relationship was not directly discussed in earlier studies on solute exchange, but can probably be attributed to the increase of turnover as stream water velocity increases, which reduces the overall HEF [18]. When turnover becomes more dominant, HEF actually declines, as shown in Figure 1 for celerities above $0.35 \, \text{m·h}^{-1}$, and exemplified by Ahmerkamp et al. [21] and Bottacin-Busolin and Marion [46].

### 4.2. Dynamics of Oxygen Patterns

The oxygen distribution pattern in the experiment with stationary bed forms matches those shown by Kaufman et al. [13] and by Precht et al. [23]. Under stationary conditions, the upwelling water through the troughs resulted in sharp transition between the anaerobic water and the stream water at the water-sediment interface (Figure 2A). Under slow and intermediate celerities ($0.04$–$0.14 \, \text{m·h}^{-1}$), advective pumping was still the dominant transport process for pore water, and the characteristic pattern of stationary bed forms, with low oxygen concentrations in chimney-like structures, are still present (Figure 2B–C and Supplementary material, Figures S1–S2: image 5–12, S3: images 14, 18–20, S4: images 3–9, Movies S1, S2, S3 and S4). The dominance of exchange due to turnover started at higher celerity ($> 0.35 \, \text{m·h}^{-1}$) where the 2D conchoidal shapes disappeared, and the oxic-anoxic interface occurred along a straight horizontal interface. This pattern was even more pronounced at the fastest celerity used in the present study ($0.67 \, \text{m·h}^{-1}$). Under these conditions, the downward transport of oxygen and other solutes is mostly limited by dispersion and diffusion processes [21]. The patterns of oxygen distribution (e.g., conchoidal shape and chimneys) are commonly visible in studies using optodes [13], and reflect well the patterns that were observed in mathematical models of transport in streambeds [21,22]. Therefore, despite the fact that we cannot strictly show that wall-effects do not exist, it may imply that the bias in advective transport is relatively small, as shown also in previous studies that used dye to trace flow patterns in bed forms [20,31,47].

The present study reveals that there is also dynamic behavior of the oxygen distribution in the sediment. For example, the offset of the redox chimneys and its relicts in the direction of the bed form stoss side during intermediate celerity of $0.14 \, \text{m·h}^{-1}$ are indicators of longer residence times in the vicinity of the points where the seepage velocity is small (Supplementary Material, Figures S3, S4 and

Movies S3, S4). The fraying of the redox chimneys was described in detail neither by the modelling studies nor by the two experimental studies of Precht et al. [23] and Ahmerkamp et al. [24]. This might be due to a lower spatial sampling and imaging resolution of the aforementioned experimental studies compared to our study. The fraying of the oxygen near the upwelling zone resulted in an irregular interface that can be explained by dispersion [47,48]. Fraying of the redox chimneys could occur also due to the dynamics between advancing or receding advective oxygen fronts and local consumption rates. In order to evaluate whether transport or reaction rates are dominating those patterns, a more detailed analysis is required. This can be done using the Damköhler number, which compares the relative rates of advective transport to reaction rates [21,24]. This can be done locally, but requires a detailed numerical modeling analysis of the flow field, which we did not conduct in this study.

Hysteresis effects on oxygenated areas upon increasing and decreasing velocity were shown by Kaufman et al. [13]. The size of the oxygenated area depends on whether the flow is accelerating or decelerating. In our case, the dynamics of the bed form movement led to different distributions of oxygen in the bed (Supplementary Material, comparison between Figures S1 and S2, S4, S5: image 1–12 and Movie S1, S2, S4, S5). Such hysteresis occurred when larger bed forms induced an advective pulse, pushing the oxic–anoxic boundary deeper into the bed, and leaving deeply-oxygenated sediment in its tail. A following, smaller bed form, having smaller advective pumping, cannot generate deep flow, and consequently cuts off deeper bed regions from oxygen resupply. The ascending oxycline can enable bed forms to reinstall temporal reconnections of deeper anoxic water with the surface (Supplementary Material, Figures and Movies S1 and S2; S4).

The complex temporal dynamics of oxygen that were observed here, as well as by Kaufman et al. [13], are expected to play an important role in redox-dependent reactions. Current modeling studies do not take into account such stochastic behavior of bed form movement, despite the fact that such statistics are well studied among geomorphologists [48]. Analyzing reaction rates in bed forms may result in underestimates when bed form dynamics are not included. For example, Kessler et al. [19] concluded in their modeling study that bed form celerity has only small effect on coupled nitrification–denitrification reactions. However, a more recent modeling study showed that both nitrification and net denitrification increase with celerity, and that both are higher under migrating bed forms. However, coupled nitrification–denitrification rates are higher under stationary bed forms. Overall, bed form migration results in a reduction of the nitrogen removal efficiency (i.e., the amount of the nitrogen that enters the hyporheic zone relative to the amount that is removed) [22]. The latter is more related to our experimental results, since it used solute levels that are more relevant to streams; in contrast, Kessler et al. [19] used conditions that were more relevant to coastal sediments. Recently, Zheng et al. [22] came up with a more detailed explanation of why different modeling studies show different patterns, and why at the moment this topic is still not fully understood because of the relatively small numbers of modeled scenarios.

### 4.3. Oxygen Consumption

The calculated volumetric uptake rates (delta method: 22–75.1 $\mu$mol L$^{-1} \cdot$h$^{-1}$, maximum uptake method: 125.1–252.4 $\mu$mol L$^{-1} \cdot$h$^{-1}$) show the efficiency of the microbial community at recycling nutrients. The rates calculated with the maximum uptake method were approximately the same as in De Falco et al. (2018), who used sediments from the same location, but used microelectrodes to measure oxygen concentrations. These rates are on the upper end of the range of respiration rates measured in marine environments (10–144 $\mu$mol L$^{-1} \cdot$h$^{-1}$), which are usually more oligotrophic than streams [24]. The oxygen uptake rates decreased monotonically with increasing celerity, and were highly correlated with flushing time. Similar trends of declining consumption rates with increasing celerity and flushing time were seen, regardless of the method that we used to calculate the absolute rates. The assumptions that all oxygen is consumed may not be valid in all systems. Indeed, marine environments with smaller reaction rates may lead to significant masses of oxygen flowing out of the bed. For example, Kessler et al. [19] showed that only ~25% of the oxygen is consumed during

flow in the subsurface. The delta method takes such behavior into account. However, the delta method assumes that all water in the hyporheic zone is transported back to the surface water with the mean concentration of the oxygenated zone. This holds true for regions with high turnover and for shallow flow paths, but not for deeper-penetrating, slower flow paths, where most or all of the oxygen is consumed. Therefore, the real respiration rates will be in the range between the values of both calculation methods. It can be concluded that the maximum uptake method is probably more correct under conditions with long residence times and high organic matter contents. The delta method more effectively represents conditions with high pore-water exchange rates.

The results of the present study suggest that it is important to consider the effects of particle mobilization and particle deposition, especially in the case of moving bed forms. For example, the respiration rates in the North Sea shelf were lower at sites affected by bed form celerity due to the wash out of organic matter from the moving fraction of the bed [24]. We observed similar patterns in the upper section of the flume sediment, which had a lower organic matter content compared to the deeper immobile sediment (Table 1; organic matter was 1.05% vs. 1.38% in the mobile and immobile sediment, respectively). The reduced amount of organic carbon may be caused by the degradation and resuspension of fine material during bed form movement. Indeed, increasing streamflow velocity and celerity led to the transfer of organic and clay particles to the surface water column, and resulted in higher turbidity and larger oxygenated zones in the bed (Table 1, Figure 4). When flow and celerity are decreased again, the organic and clay particles are deposited, and may shift the flow paths in the bed due to clogging, or may create active zones due to the concentration of organic matter. The deposition varies with velocity and celerity, and can be located near the sediment–water interface near the inflow zone [49], or at the interface between the mobile-immobile fractions of the bed. Such sorting and deposition processes may greatly influence the fluxes due to clogging or reaction rates because of the flushing of organic matter from the reactive zones [50].

The majority of hyporheic studies that focus on biogeochemistry and ecology were conducted under stagnant bed conditions. In contrast, studies on the morphology of streambeds included the motion of the sediments. However, the connection between these topics remains loose, despite the fact that bed migration dramatically changes the physical environment in the bed. The results presented in this study imply that it is essential to incorporate the processes affected by bed form migration into future modeling of stream bed processes, with implications for river network modeling [51,52]. All biogeochemical processes are affected by the flux of nutrients from the water into the sediments, where most of the microbial activity occurs. The zone that is affected by HEF is also critical for process quantification, since the volume of sediment that is involved in a reaction dictates also the amount of biomass that is active (e.g., for oxidation). We have shown that within the hyporheic zone, different flow conditions can result in complex interfaces between oxic and anaerobic zonation, which can lead to variable volumetric reaction rates due to the changing fluxes [22]. While modeling of the aforementioned processes is essential for understanding process coupling, special efforts are needed to incorporate other processes that are known to affect biogeochemical processes but which are currently neglected. These include, for example, fine particle transport and deposition, the physical impacts of sediment motion on biofilms, etc.

## 5. Conclusions

This study has clearly shown, in a series of highly-controlled flume experiments, that bed form celerity has a significant impact on the dynamics of oxygen distribution and uptake rates in the HZ. To the best of our knowledge, this is the first study that shows high spatial and temporal resolution of two-dimensional oxygen distribution during bed form migration in a freshwater system. The transport of oxygen into the bed was purely advection-dominated in a stationary bed, but gradually changed into a turnover-dominated system as celerity increases. This resulted in an increase in the streambed volume that was exposed to oxygen, but not necessarily in higher oxygen fluxes into the sediment. However, because oxygen uptake depends on the combination between the volume of the oxygenated

bed, biomass, and oxygen fluxes, increasing celerity resulted in a reduction in the average volumetric oxygen uptake rate. Part of this was due to the washout and deposition of fine particles, including organic material, and due to a reduction in the flushing times of the hyporheic zone.

**Supplementary Materials:** The following are available online at http://www.mdpi.com/2073-4441/12/1/62/s1, Figure S1: Oxygen distribution during bed form celerity of 0.04 m·h$^{-1}$ (run 3). Images from left to right, Figure S2: Oxygen distribution during bed form celerity of 0.04 m·h$^{-1}$ (run 8). Images from left to right, Figure S3: Oxygen distribution during bed form celerity of 0.14 m·h$^{-1}$ (run 4). Images from left to right, Figure S4: Oxygen distribution during bed form celerity of 0.14 m·h$^{-1}$ (run 9). Images from left to right, Figure S5: Oxygen distribution during bed form celerity of 0.35 m·h$^{-1}$ (run 2). Images from left to right, Figure S6: Oxygen distribution during bed form celerity 0.35 m·h$^{-1}$ (run 5). Images from left to right, Figure S7: Oxygen distribution during bed form celerity of 0.35 m·h$^{-1}$ (run 10). Images from left to right, Figure S8: Oxygen distribution during bed form celerity of 0.67 m·h$^{-1}$ (run 6). Images from left to right. Figure S9: Oxygen distribution during bed form celerity of 0.67 m·h$^{-1}$ (run 11). Images from left to right. Movie S1: Run 3, Movie S2: Run 8, Movie S3: Run 4, Movie S4: Run 9, Movie S5: Run 2, Movie S6: Run 5, Movie S7: Run 10, Movie S8: Run 6, Movie S9: Run 11

**Author Contributions:** Conceptualization, P.W., S.A. and J.L.; methodology, P.W., and C.D.; software, P.W. and Y.T.; formal analysis, P.W., C.D., and Y.T.; data curation, P.W; writing—original draft preparation, P.W., and S.A.; writing—review and editing, all authors; funding acquisition, S.A, P.W., and J.L. All authors have read and agreed to the published version of the manuscript.

**Funding:** This research was supported by the Israel Science Foundation (grant 682/17) and by the Young Scientist Exchange Program of the German Federal Ministry of Education and Research (BMBF) and the Israeli Ministry of Science, Technology and Space (MOST) in the framework of the German-Israeli Cooperation in Water Technology Research (grant number YSEP-122).

**Acknowledgments:** We thank Soeren Ahmerkamp for assistance with modeling calculations.

**Conflicts of Interest:** The authors declare no conflict of interest.

**Appendix A**

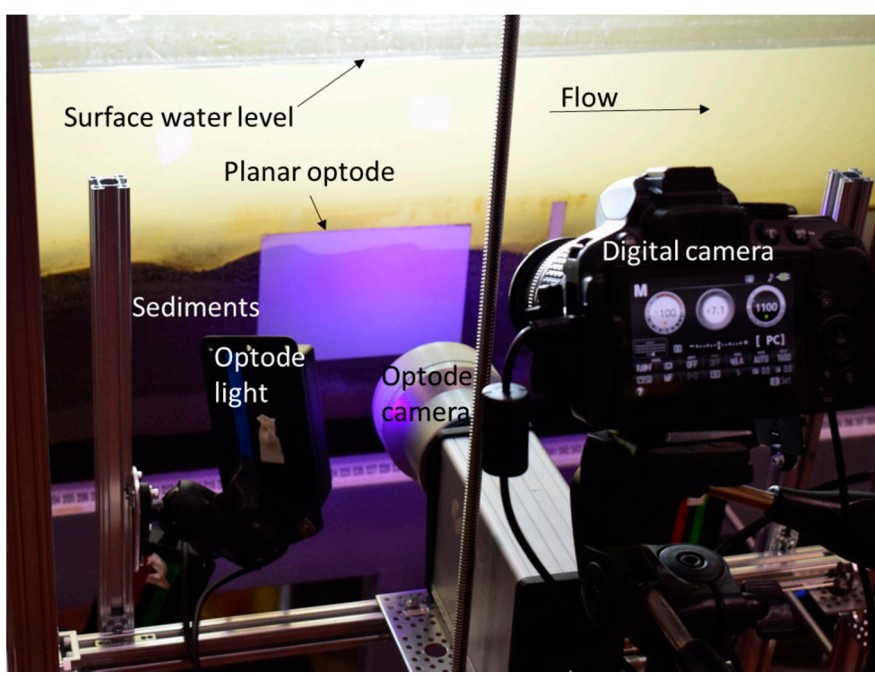

**Figure A1.** Experimental set-up of the planar optodes.

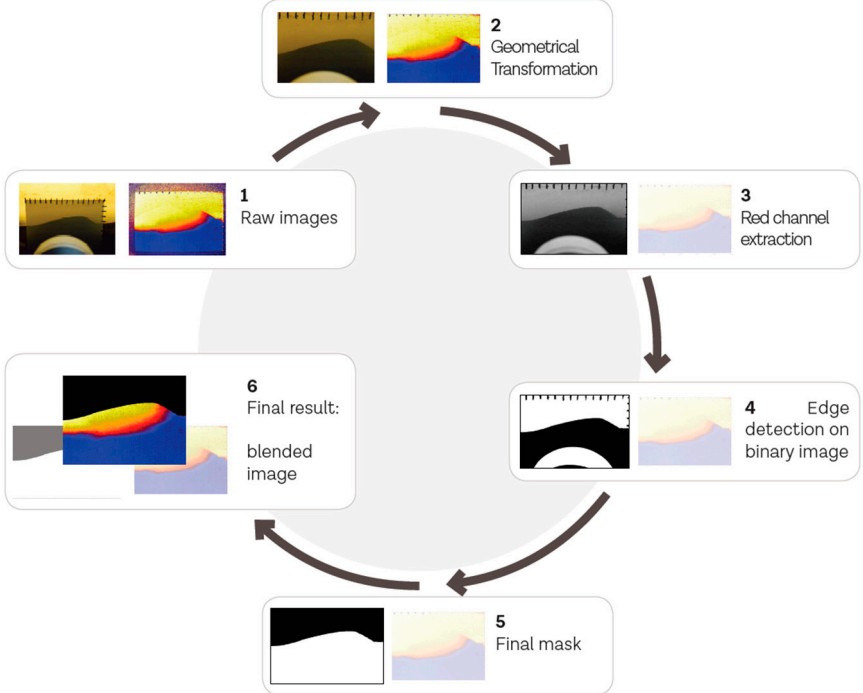

**Figure A2.** Flow diagram showing the stepwise image processing procedure to merge the topography information from a digital image with the spatial oxygen saturation information from the optode into one image.

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
