# Peer review of "Impact of Bed Form Celerity on Oxygen Dynamics in the Hyporheic Zone"

_water, doi:10.3390/w12010062_

Round 1
Reviewer 1 Report
The manuscript by Wolke et al. investigates the role of bedform migration on oxygen exchange through the sediment-water interface of permeable sediments. The authors collected a unique dataset of oxygen distributions underneath stationary and migrating bedforms obtained from flume experiments with reactive sediments. The presented parameter variations, i.e. variable flow speeds and bedform celerities, allow for new insights into the biogeochemical processes within riverbeds and coastal sediments.
Overall, the manuscript is technically sound, well written and thoughtfully structured; however it gives away some of its potential by a superficial analysis of the dataset and a discussion that can be strengthen. In the following review, I will address these points in more detail with recommendations to improve the manuscript. In my opinion, the major issues can readily be addressed and after a careful revision, the manuscript is worth its publication and it will find a broad readership from different research fields.
Major Comments:
It is not clear whether the authors assume that their flume-system is in stationary or instationary state. To my knowledge, all of the cited modelling studies are based on the assumption of stationarity even when considering bedform migration (Ahmerkamp et al., Bottacin and Busolin et al, etc take advantage of using a coordinate transformation). Within the dataset, Wolke et al. see clear indications that stationarity assumption is not necessarily valid: “It was observed that bed form celerity under the same flow conditions can be temporally variable” (Line 192). I am convinced that this is not only a laboratory artifact but is also occurring under field conditions. The implications are very important and interesting, as the instationarity would decouple the concurring effects of advective pumping and turnover by bedforms, thereby the total oxidized volume is likely enhanced (not necessarily fluxes). I am aware that this dynamic effects are qualitatively discussed (Line 314-Line 322), however based on the obtained results these effects can actually be quantified which could be a very valuable contribution to the manuscript. The simplest approach would be to calculate the difference of two sequential oxygen distribution to estimate rates of change. These rates of change depend on advection and the volumetric rates and would allow for some interesting quantifications. Volumetric rate measurements: The authors use a relatively complicated method to determine volumetric oxygen consumption rates based on fluxes, in which the flux estimates underlie several assumptions. Volumetric rate measurements are one of the easiest measures as they could be inferred from flow-through reactors (e.g. Rao et al.), through point measurements using oxygen probes (e.g. Polerecky et al.) and even from the planar optodes (e.g. Kessler et al. 2013). Why did the others not use one of the conventional methods? I am disagreeing to the statement “Direct comparison between experimental and modeling studies is somewhat difficult” (Line 349) and actually, I am missing a more in depth comparison to modelling results. Bottacin-Busolin and Marion, Ahmerkamp et al. and Elliott and Brooks are all providing simple scaling laws (at least for stationary bedforms), which allow for the prediction of exchange fluxes induced by advective pumping. Wolke et al. measured the right parameters (hydraulic conductivity, flow velocity, oxygen penetration depth, etc) on which basis exchange fluxes could be estimated and added to figure 1 and 4. I strongly recommend adding this comparison to the manuscript as it will give the reader a better feeling on predictability of exchange fluxes and errors of the models. In short, based on the presented parameters a theoretical characteristic porewater velocity can be estimated (Elliott and Brooks), which can then be compared to reaction timescales (keyword: Damköhler number, Ahmerkamp et al.), bedform celerity (Bottacin-Busolin and Marion) and hyporheic exchange flow. In addition, the onset of bedform migration can easily be compare to existing empirical relationship. Boundary effects and planar optodes: I agree that planar optodes are currently the best available method to visualize high-resolution 2-dimensional oxygen distributions. However, the technique comes with some limitations, such as a bias by wall-effects. Do the authors assume that the oxygen distributions in the center-section are similar to the ones measured close to the wall? Is the advection inhibited? Is the hydraulic conductivity affected? These limitations have to be addressed in the manuscript. I really enjoy reading about the effects of organic carbon, however in the discussion the strongly coupled effects of organic carbon content, turbidity and volumetric reaction rates are disconnected. The effect of decreasing volumetric reaction rates with increasing bedform celerity can directly be linked to the decreased organic carbon stock in the sediments and that should clearly be stated.
Minor Comments:
Line 34: HEF is sometimes used for hyporheic exchange flux (e.g. line 34) and sometimes hyporheic exchange flow (e.g. Figure 1) which have different units, please be consistent with terminology throughout the manuscript. My personal opinion is that abbreviations decrease the readability and I would suggesting to reduce the amount or to not use them at all – however I leave this to the authors.
Line 177: Was the turbidity constant after each individual run? Did the total turbidity result from the total duration of the experiments, or was it indeed changing between the different flow speeds? Were the turbidity effects reversible?
Line 220: Cool, I would like to see this temporal dynamics, see comment below.
Line 240: Oxygen fluxes are not shown at all? Please add those to the manuscript.
Line 307: Why should denitrification not occur in deeper sediment layers?
Line 326: This statement is wrong - to which publication are the authors referring to? The Kessler et al. (2015) paper was proven wrong by several follow-up publications and the general understanding is that - similar to the oxygen fluxes - also nitrification and denitrification decline with increasing bedform celerity (see Ahmerkamp et al. and Zheng et al.)
Line 337: It would be helpful if the mentioned rates were also shown in brackets.
Line 372-375: The sentence is not clear to me. Do the authors expect that the microbial communities only live in the oxic zone or the upper sediment layers? There will be rates in all sediment layers. In Zheng et al. it was not shown that the rates change, but fluxes.
Line 389: There are at least two other studies among the cited ones that also show high resolution oxygen distributions underneath stationary and migrating bedforms.
Methods and Figures:
Line 87: Do the authors assume that the first 10 cm of sediment are regularly oxidized? How did the authors ensure that reduced substances and subsequent oxidization were not affecting the volumetric rate estimates?
Line 116-117: The arithmetic mean between trough and crest should be 0!? Why not using standard methods like 2*sqrt(2)*standard_deviation?
Line 120: Which lense was used to observe the bedforms? Why did the authors use two different systems for bedform height (acoustic bottom track) and bedform celerity (camera system)? Are the determined bedform heights and celerities similar for both methods?
Line 126: As HEF is an important measure, I would like to read more about the calculation method.
Line 133: What is the excitation, emission wavelength of the optode system?
Equation 2-7: I recommend to use “⋅” instead of “*” as “*” is usually used for the convolution of two matrices. Some of the described parameters, like the oxygen concentration, are 2-d matrices and I guess the authors just used pointwise multiplications.
Eq. 3-5 and Line 168: It is not clear how q_h is calculated.
Eq. 3: I find the term “total uptake method” misleading as the equation describes the calculation of a volumetric reaction rate and “total uptake” usually refers to net fluxes.
Line 185: How was the porosity calculated?
Line 188: How was the organic carbon content calculated?
Figures: The figures could benefit from some improvements: a) the supplementary figures should have the same color-coding as figure 2. Why is the oxygen saturation going below 0 and far above 100 % (?), it seems that the calibration is off. The authors have measured the oxygen concentration in the water column (100%) and in the deep sediment (0%), thereby each image could be recalibrated which would significantly improve the accuracy and resolution (please notice that the relationship between the ratiometric measurement and oxygen concentration is not linear). The black background in figure 2 should be removed from the colorbar and the font should be adapted to the other figures. B) In figure 1, 3 and 4 the fonts are too small and in a black white print, it is not possible to distinguish the symbols within figure 3 and 4. C) Figure 4 linear fit: Would you expect a linear relationship and I think it is sufficient to only show A and B.
Supplementary Figures: I recommend compiling the time series of oxygen distributions into a gif or video file, as this would help understanding the temporal dynamics.
Bibliography
Ahmerkamp, Soeren, Christian Winter, Felix Janssen, Marcel MM Kuypers, and Moritz Holtappels. "The impact of bedform migration on benthic oxygen fluxes." Journal of Geophysical Research: Biogeosciences 120, no. 11 (2015): 2229-2242.
Bottacin‐Busolin, Andrea, and Andrea Marion. "Combined role of advective pumping and mechanical dispersion on time scales of bed form–induced hyporheic exchange." Water resources research 46, no. 8 (2010).
Elliott, Alexander H., and Norman H. Brooks. "Transfer of nonsorbing solutes to a streambed with bed forms: Theory." Water Resources Research 33, no. 1 (1997): 123-136.
Kessler, Adam J., Ronnie N. Glud, M. Bayani Cardenas, and Perran LM Cook. "Transport zonation limits coupled nitrification-denitrification in permeable sediments." Environmental science & technology 47, no. 23 (2013): 13404-13411.
Kessler, Adam J., M. Bayani Cardenas, and Perran LM Cook. "The negligible effect of bed form migration on denitrification in hyporheic zones of permeable sediments." Journal of Geophysical Research: Biogeosciences 120, no. 3 (2015): 538-548.
Polerecky, Lubos, Ulrich Franke, Ursula Werner, Björn Grunwald, and Dirk de Beer. "High spatial resolution measurement of oxygen consumption rates in permeable sediments." Limnology and Oceanography: Methods 3, no. 2 (2005): 75-85.
Rao, Alexandra MF, Mark J. McCarthy, Wayne S. Gardner, and Richard A. Jahnke. "Respiration and denitrification in permeable continental shelf deposits on the South Atlantic Bight: N2: Ar and isotope pairing measurements in sediment column experiments." Continental Shelf Research 28, no. 4-5 (2008): 602-613.
Author Response
Response appear in the attached file

Reviewer 2 Report
This manuscript presents the results of an experimental activity aimed at studying hyporheic exchange and oxygen consumption in the hyporheic zone created by moving dunes. The experiments and the interpretation are two-dimensional.
General comments
I am in general in favor to experimental activities and this one is well organized in my view. What is laking is a clear and solid interpretation of the experimental results. For instance, I am not fully convinced of the explanation of the different oxygen distribution emerging as the stream velocity increases. Firstly, while the change in the bed form is evident nothing is said about water depth and stream hydrodynamics, which is certainly a driver of the observed change. The change in oxygen dynamics can be simply explained by the increase of the HEF with stream water velocity, which is associated to a reduction of the residence time.
The sharper interface between oxic and anoxic areas may be due to the compression of streamlines to the water-sediment interface, in turn triggered to the change in dune geometry, which leads to a larger horizontal component of the velocity at the interface with respect to the the small stream velocity case. In principle, this may be explained by the same advective model proposed for the low celerity case, and it is unclear to me why you seem to esclude this model in the discussion from line 282 to line 292.
The following discussion (lines 293-300) is unclear to me. I am not sure that what described here can be considered a real stagnation point. If I understand correctly the situation you are describing here is that of a seepage velocity equal, or close, to the bed form celerity. Instead a stagnation point is a point in which seepage velocity is zero. As written this interpretation seems to suggest that bed forms migrates by rigid translation, not by a wave type of change induced by sediment transport interesting the interface. I am not a morphologist and I may be wrong but the interpretation you proposed here is confusing me. Instead, the offset of the anoxic chimney may be explained by the change in the shape of the bedroom and the consequent change in the hyporheic velocity and the residence time, again controlled by advection. In the following you enter into detailed discussions, which are hard to follow because the reference to the figures in the supplementary material is not focused. In other words, it is unclear to what particular figure and panel of a large number of figures you are referring when discussing single aspects of the results. This makes life of the reviewer, and I think also of the reader, hard and the message becomes unclear.
In my view the manuscript requires moderate revision, chiefly in the discussion section, aimed at strengthening the physical interpretation of the experimental results.
Author Response
Response is attached as a file

Round 2
Reviewer 1 Report
Wolke et al. adequately addressed my comments and I enjoyed reading the revised version of the manuscript, however I am left with two major and a few minor comments. After the authors addressed these comments, the manuscript is ready for publication.
Major comment
In Line 376-380 the authors are stating that "the Damköhler [...] number will not assist in understanding the local dynamics of oxygen". This is a very strong statement which is not correct. Ahmerkamp et al. 2017 investigated oxygen penetration depths for 12 sandy stations in the North Sea - all of the stations had different grain sizes / flow conditions / bedform celerities. Overall, the Damköhler-scaling was surpringsly well predicting average oxygen fluxes (Fig 9 in Ahmerkamp et al. 2017). In addition, the variability of the oxygen penetration depth along a tidal cycle was correlated with the Damköhler number (Fig 7 in Ahmerkamp et al. 2017). The non-dimensional Damköhler number does not cover all of the dynamics to the smallest detail but it is a good indicator for average oxygen penetration depths and its variability. Nitrogen cycling and migrating bedforms: In the revised version, the authors are still stating that "nitrification and denitrification increase with celerity, and that bed form migration results in higher rates of nutrient cycling as compared to stationary bed forms". This is not correct and I am disagreeing with this statement at least in the way it is currently written up. The authors themselves are showing that bedform celerity has a negative impact on oxygen rates obviously that will also apply for n-cycling rates as a) it is the same bacteria that respire oxygen and switch their pathways to nitrate if oxygen depletion occurs, b) also for denitrification organic carbon is needed, if organic carbon gets washed out it will negatively affect denitrification, c) a lot of the ammonium required for nitrification comes from oxygen driven remineralization. Citation from the abstract of Zheng et al.: " The nitrate removal efficiency increased asymptotically with Damköhler number for both mobile and immobile ripples, but the immobile ripple always had a higher nitrate removal efficiency."Minor comments
The authors should briefly describe the Elliott model in the methods section Figure 1: The scaling of the axis should be changed so that the variability of the exchange flux is emphasized. Line 406: Mention in brackets the range of measured volumetric rates Line 410: I guess it is µmol l-1 h-1 and not mmol l-1 h-1, otherwise the rates would be gigantic.Comment to authors that does not need be adressed:
Line 414-424: Indeed, there are many systems in which not all of the oxygen is consumed within the sediment and a large portion is recirculated into the water column. This is nicely shown by the scaling law of Elliott et al. which was extended by Ahmerkamp et al. for reactive systems (by the integration of the Damköhler number). The scaling law follows a logarithmic shape. For large volumetric rates the log-function can be approximated by a linear trend, i.e. all oxygen is consumed in the vicinity of the sediment surface, at some point the curve starts to level off (oxygen penetrates deeper) which results from the increasing amounts of water+oxygen that are recirculated into the water column.
Author Response
Response to reviewers is attached
